# Needs- and user-oriented development of contactless camera-based telemonitoring in heart disease–Results of an acceptance survey from the Home-based Healthcare Project (feasibility project)

Peggy Borchers[1][*], David Pfisterer[1], Matthieu Scherpf[2], Karen Voigt[1], Antje Bergmann[1]

1 Faculty of Medicine Carl Gustav Carus, Department of General Practice, Medical Clinic III, Technische Universität Dresden, Dresden, Germany, 2 Institute for Biomedical Engineering, Technische Universität Dresden, Dresden, Germany

☯ These authors contributed equally to this work.

* Peggy.Borchers@uniklinikum-dresden.de

**Data Availability Statement:** All data are in the paper and its Supporting Information files.

## Abstract

Home-based telemonitoring in heart failure patients can reduce all-cause mortality and the relative risk of heart failure-related hospitalization compared to standard care. However, technology use depends, among other things, on user acceptance, making it important to include potential users early in development. In a home-based healthcare project (a feasibility project) a participatory approach was chosen in preparation for future development of contactless camera-based telemonitoring in heart disease patients. The project study patients (n = 18) were surveyed regarding acceptance and design expectations, and acceptance-enhancing measures and design suggestions were then drawn from the results. The study patients corresponded to the target group of potential future users. 83% of respondents showed high acceptance. 17% of those surveyed were more skeptical with moderate or low acceptance. The latter were female, mostly living alone, and without technical expertise. Low acceptance was associated with a higher expectation of effort and lower perception of self-efficacy and lower integratability into daily rhythms. For the design, the respondents found independent operation of the technology very important. Furthermore, concerns were expressed about the new measuring technology, e.g., anxiety about constant surveillance. The acceptance of a new generation of medical technology (contactless camera-based measuring technology) for telemonitoring is already quite high in the surveyed group of older users (60+). Specific user expectations concerning design should be considered during development to increase acceptance by potential users even more.

**Funding:** The Home-based Healthcare Project received funding from the European Regional Development Fund and the Free State of Saxony (grant number: 100278533). The funders had no role in study design, data collection and analysis, decision to publish, or preparation of the manuscript.

**Competing interests:** The authors declare they have no competing interests.

## Introduction

For decades the diagnosis of heart failure in Germany has led to more hospitalizations than any other diagnosis [1] and with an upward trend [2]. This has a relevance to healthcare policy and economics that is prompting the development of numerous telemonitoring applications (TMA) for heart failure. The aim of TMAs is improved monitoring and early detection of medical situations to enable timely medical intervention and thus prevent disease escalation and hospitalization. The resulting reduction in hospitalizations is a frequent outcome that has been investigated in many studies of telemonitoring interventions in heart failure patients [3–5]. A comprehensive overview of reviews from 2015 on the effects of home-based telemonitoring interventions in patients with chronic heart failure shows that home-based telemonitoring interventions reduce the relative risk of all-cause mortality and heart failure-related hospitalization compared to usual care (risk reduction between 1.4% and 8.2%) [6]. More recent studies also show significant reductions in all-cause mortality, reduction in hospitalizations due to cardiac decompensation [7] respectively a reduction in unplanned hospitalizations due to heart failure in subgroups [8]. In Germany, telemonitoring as a digital form of care for patients with heart failure was recognized as an independent examination and treatment method in 2021 by the decision of the Federal Joint Committee (G-BA) and is therefore reimbursed as a standard service [9]. This decision applies to both invasive and non-invasive telemonitoring.

However, telemedicine studies also show that the use of telemonitoring applications is dependent, among other things, on user acceptance [10,11]. It is therefore important to include potential users early in the development process [12]. It is especially important, precisely for heart failure patients who are often elderly, to consider the age-specific aspects of cognition, perception and behavior in the user-centered design approach to telemedicine applications [13]. Just as in the development of health and medical apps, user expectations [14] or acceptance prior to actual use [11], and the needs of potential future users should be identified a priori in order to include them in the development [14]. Implementation strategies can then be developed to enable elderly patients (>70 years) to quickly adjust to the telemonitoring and integrate it into their daily lives and routines [15].

This type of participatory approach was taken in the Home-based Healthcare Project (*Projekt Häusliche Gesundheitsstation*). To do this, the needs and acceptance of potential users were identified a priori so as to include them in the development of a new generation of patient monitoring (contactless and camera-based).

This was done in the context of reducing future barriers to users and thus clearly increase patient acceptance and comfort in regard to the planned TMA. An acceptance survey was administered to the study patients and study physicians participating in the Home-based Healthcare Project to gather data on their acceptance prior to actual use and needs. The results form the focus of this paper.

## Methods

### 1. Home-based Healthcare Project

The Home-based Healthcare Project, underway since August 2018, is a telemedicine feasibility project funded by the European Regional Development Fund and the Free State of Saxony (grant number: 100278533). The overarching project goal is the development of an innovative system to be used in primary care as a means of monitoring in chronic disease patients with elevated cardiovascular risks (e.g., heart failure). This innovative system is based on a new generation of medical technology, namely a camera-based contactless measuring technology. This project focuses on the feasibility and evaluation of the measuring technology.

The core technical component is imaging photoplethysmography (iPPG), which has been undergoing development since 2012 at the Institute for Biomedical Engineering, as well as at other locations [16–19]. It enables the measurement of cardiovascular signals such as heart rate and respiration rate, as well as the estimation of oxygen saturation and blood pressure, with the cameras in standard tablets or smartphones [20].

In one extensive work package of the Home-based Healthcare Project, the use of this camera-based contactless measuring technology was investigated by an evaluation study in the ambulant primary care setting. This entailed a) the development of an appropriate measuring station, b) the application of extant algorithms to extract the vital parameters from the camera data, and c) the evaluation of the contactlessly measured vital parameters.

The acceptance survey focused on the further development of this measuring technology into a TMA and was conducted as part of the evaluation study using the participating study patients.

This project, including the evaluation study and acceptance survey, was reviewed by the Ethics Commission at the Technische Universität Dresden and approved (reference number: EK 412092019). All participants received written study informations and were informed about the study verbally by their primary care physician. All participants gave their written informed consent. The study was carried out in accordance with the Declaration of Helsinki.

## 2. Evaluation study in the Home-based Healthcare Project

The primary aim of the evaluation study was to investigate the precision and reliability of the contactlessly measured vital parameters compared to vital parameters measured in the usual manner. To do this, synchronous evaluation measurements were taken in the study patients using a tablet camera and reference measuring devices. Measured with standard measuring devices were pulse and blood pressure (oscillometric blood pressure device), oxygen saturation (Spo2 finger and ear clip) and respiratory rate (portable Philips IntelliVue X3 monitor).

The evaluation study was conducted in close collaboration between the Institute for Biomedical Engineering at the Technische Universität Dresden (development of the measuring station and analysis of the measurements) and the Department of General Practice at the Technische Universität Dresden (recruitment and taking evaluation measurements).

From January 2020 to July 2021, nineteen of the initially planned 20 study patients were included in the evaluation study and 250 evaluation measurements (an average of 13.16 per study patient) were taken in the homes of the study patients by study nurses. Unfortunately, the inclusion of the 20 planned study patients could not be fully achieved despite intensive efforts because the corona pandemic occurred during the recruitment period and made it very difficult to recruit the corresponding patient group. Inclusion criteria was, in addition to a minimum age of 18 years, the presence of one of the following diagnoses: chronic heart failure (at least NYHA stage 2), coronary artery disease, hypertensive heart disease, chronic obstructive pulmonary disease (COPD, at least gold stage 3) or aortic valve stenosis (at least moderate). For a complete list of inclusion and exclusion criteria, see S4 Table. Since the evaluation study was similar in character to a pilot study, case number planning was based on pragmatic reasoning.

## 3. Acceptance survey regarding further development of an iPPG-based TMA

The acceptance survey was already oriented toward the next step in the prospective development of this measuring technology into a TMA for independent use by patients. This involves a planned TMA undergoing development based on the results of the evaluation study.

**Goals of the acceptance survey.** The goals of the acceptance survey were to determine a priori a) user acceptance prior to actual use, b) expectations regarding design, and c) concerns of potential future users (physicians and patients) regarding the prospective TMA and, based on that, to identify acceptance-enhancing measures and design suggestions for consideration during technical development.

The acceptance survey consisted of questionnaires administered to the study patients in the evaluation study and qualitative interviews held with the study physicians. The results of the study patient survey are presented in this paper.

**Development of the questionnaire.** The questionnaire (S1 Fig–in German and English) is based on the Unified Theory of Acceptance and Use of Technology (UTAUT) proposed by Venkatesh et al [21]. Corresponding to Venkatesh et al. acceptance of a technology aims at the behavioral intention, which in turn has an influence on use behavior [21,22].

However, in our case, this entails a planned TMA, so we refer in this study to the intention to use the technology.

The questions were taken from the separate acceptance determinants in the UTAUT model, translated and formulated in the subjunctive, and adapted to fit the planned TMA. The determinants of effort expectancy, performance expectancy and social influence were taken from the UTAUT model. The facilitating conditions were not included since the model put forth by Venkatesh et al. shows that these only have an influence on actual use (Fig 1).

The determinants self-efficacy, attitude toward using technology, and anxiety (items used in estimating UTAUT) were also included because these determinants are important for the targeted user group since habits play a particularly central role for elderly patients and therefore should be given special attention. For instance, a TMA should be inserted as unobtrusively as possible into the usual daily routine to avoid placing excessive demands on the patients [23].

For these reasons two additional determinants of our own were included on the questionnaire with corresponding questions. These are the integrability into the daily routine and the influence of the technology on the user. The questions formulated for this are found in S1 Fig (in German and English).

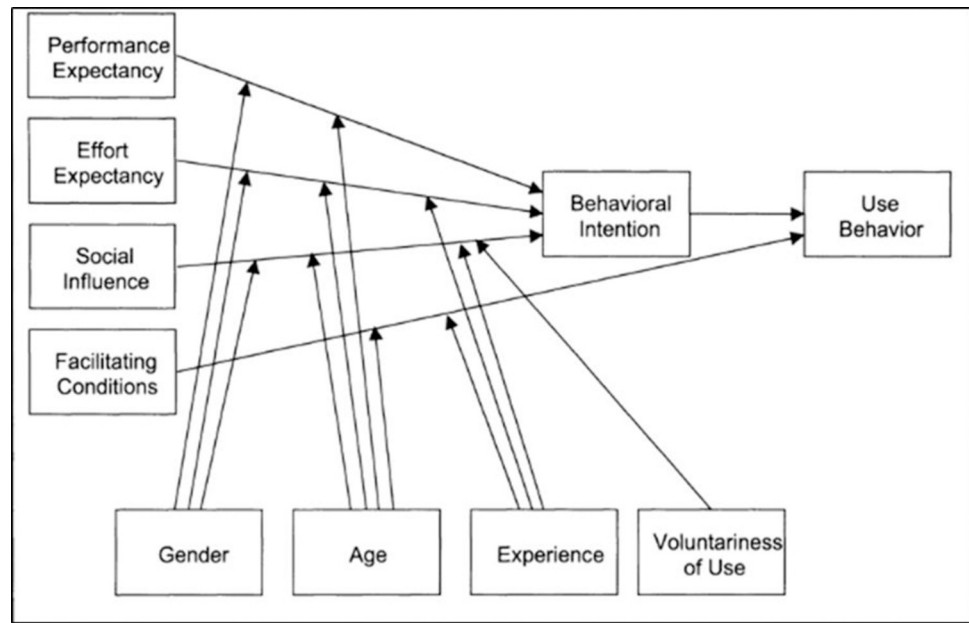

**Fig 1. UTAUT model (Venkatesh et al., 2003) [21].**

The questions about the determinants were asked on a six-point Likert scale with the options ranging from "completely agree" to "completely disagree." The questionnaire was also expanded to include questions about sociodemographic issues and general attitude toward technology, as well as qualitative questions not only focused specifically on the patients' expectations regarding the design of the planned TMA, but also on any possible concerns.

**Data collection.** The surveys took place between May 2020 and December 2020, after several evaluation measurements had already been carried out on the study patients and they were familiar with the measurement technology. Due to the onset of the corona pandemic in spring 2020, home visits to patients were not possible for a while. Therefore the surveys were conducted orally via telephone by study nurses using the structured questionnaire, not as originally intended in writing and at the end of the evaluation study. To ensure comparability of the results, the conduction via telephone (after several evaluation measurements) was retained for all participants until the end of the study. The responses were recorded by hand on paper questionnaires. Participation in the survey was voluntary.

**Analysis.** The recorded survey responses were coded appropriately and entered in the statistics software SPSS (version 27). The analysis is descriptive taking differences between the individual groups into consideration.

To address the acceptance more concretely, we developed our own categorization of the acceptance level for the intended use by dividing it into three categories (high, moderate, low) (S1 Table). To be able to make claims about estimations of the determinants, each determinant was likewise categorized into three groups, analogous to the categorization of the acceptance level (S2 Table). As a result, it was possible, for instance, to estimate the performance expectancy for each respondent as "high," "moderate" or "low." The mean value was used to categorize determinants with multiple items. An example of this is presented in S3 Table.

By categorizing in this way, the analysis enabled conclusions about different groups (e.g., groups with high acceptance) and the identification of potential differences in their determinant values because it was primarily the values and attitudes of the group with low/moderate acceptance that were important for identifying acceptance-enhancing factors.

## Results

Eighteen out of 19 study patients in the evaluation study participated in the telephone interviews. One study patient declined to take part in the interview due to deterioration in health. Prior to the interview, an average of 5 evaluation measurements were carried out at the patients' homes. The age ranged between 68 and 101 years. 61% (n = 11) were female, 56% (n = 10) lived in a partnership and 67% (n = 12) stated that they had a basic interest in technology. The participants' sociodemographic characteristics and general attitudes toward technology are presented in Table 1.

### a) User acceptance

Fifteen of the 18 (83%) respondents agreed with an intention to use the planned TMA (completely agree + agree). According to the categorization their acceptance is high. They are referred to in the following as Group 1, "high acceptance. "Three respondents disagreed (n = 1) or mostly disagreed (n = 2) with an intention to use the planned TMA. Their acceptance is moderate or low according to the categorization, summarized below in group 2 "moderate/low acceptance".

**Comparison of the sociodemographic characteristics and general attitudes toward technology between the groups.** The age of those in Group 2 was higher in terms of mean and median than in Group 1 and in the entire group. The respondents in Group 2 were female.

**Table 1. Presentation of the respondents' sociodemographic characteristics and general attitudes toward technology, as well as of the groups with high acceptance vs. moderate/low acceptance.**

| | All respondents (n = 18) | | Group 1 high acceptance (n = 15) | | Group 2 low/moderate acceptance (n = 3) | |
|---|---|---|---|---|---|---|
| **Age (in years)** | | | | | | |
| Minimum | 68 | | 68 | | 78 | |
| Maximum | 101 | | 93 | | 101 | |
| Mean | 81.9 | | 81 | | 89 | |
| Median | 84 | | 83 | | 88 | |
| | Absolute frequency (n) | Relative frequency (%) | Absolute frequency (n) | Relative frequency (%) | Absolute frequency (n) | Relative frequency (%) |
| **Sex** | | | | | | |
| Female | 11 | 61% | 8 | 53% | **3** | **100%** |
| Male | 7 | 39% | 7 | 47% | 0 | 0% |
| **Family status** | | | | | | |
| Living alone | 8 | 44% | 6 | 40% | **2** | **67%** |
| Living with others | 10 | 56% | 9 | 60% | 1 | 33% |
| **Interest in technology** | | | | | | |
| Yes | 12 | 67% | 11 | 73% | 1 | 33% |
| No | 6 | 33% | 4 | 27% | **2** | **67%** |
| **Experience with technology**[*] | | | | | | |
| High | 4 | 22% | 4 | 27% | 0 | 0% |
| Moderate | 9 | 50% | 8 | 53% | 1 | 33% |
| Low | 5 | 28% | 3 | 20% | **2** | **67%** |
| **Presence of modern media/technology** | | | | | | |
| Yes | 14 | 78% | 13 | 87% | 1 | 33% |
| No | 4 | 22% | 2 | 13% | **2** | **67%** |

[*]Regarding the operationalization of experience with technology, see questionnaire S1 Fig).

Compared to Group 1 they were more likely to live alone, have mostly no interest in technology and no experience with technology, and do not possess any modern technology (see Table 1).

**Comparison of the acceptance determinants between the groups.** Differences were visible between the groups for all determinants except social influence and anxiety (see Table 2). The most clearly visible differences were seen for the determinants effort expectancy, integrability into daily routine, and self-efficacy. Group 2 estimated the expected effort to be higher and the integrability and their self-efficacy to be lower than Group 1 did. In this context, low self-efficacy means that Group 2 had difficulty imagining independent use even with an integrated help and/or support function.

Fig 2 shows an illustration of the attributes for moderate/low acceptance.

## b) Patients' ideas about the design of the planned TMA

The two groups were considered together in regard to this. A total of 16 (89%) of the respondents stated that the camera should not be constantly on and it is important that the camera can be actively switched on and off for the measurements.

**Table 2. Determinant values in the case of high acceptance vs. moderate/low acceptance.**

| | Group 1 High acceptance (n = 15) | Group 2 Moderate/low acceptance (n = 3) |
|---|---|---|
| **Determinant** | Categorization (MV)* | Categorization (MV)* |
| Performance expectancy | high (5.4) | moderate-high (4.5) |
| Effort expectancy[1] | low (5.1) | moderate (4.2) |
| Social influence | high (5.3) | high (5.3) |
| Attitude toward technology | positive (5.2) | neutral-positive (4.5) |
| Self-efficacy | high (4.8) | moderate (3.2) |
| Anxiety | moderate (2.6) | moderate (3.3) |
| Integratability into daily routine | high (5.2) | moderate (3.0) |
| Influence of technology | moderate (3.4) | moderate (3.0) |

*This information is based on the calculation of the mean values of the determinants for the individual respondents (values between 1–6).

[1]The determinant effort expectancy is set as a polar opposite.

Fourteen (78%) of the respondents preferred direct feedback concerning the measured values. Of these, half (n = 7) favored visual feedback on the device into which the camera would later be integrated. The other half favored acoustic or combined (visual and acoustic) feedback. Multiple answers were possible.

Eleven (61%) of the respondents desired that the measured values be regularly transmitted to their primary care physicians. Four (22%) found such a transmission good only if the physician also desired this or transmission only took place if abnormal vital parameters were measured. Two of the respondents stated concerns that the sheer volume of transmitted values could potentially overwhelm the primary care physician.

The option of having a camera in a fixed location was also described. In respect to this aspect, the respondents were most frequently able to imagine having the camera in the bathroom (33%) or living room (28%), followed by the study-room (17%) and the kitchen (11%). The bedroom was mentioned only once as a potential place for the camera. Furthermore, two of the respondents (11%) categorically rejected the idea of a camera in the bedroom or

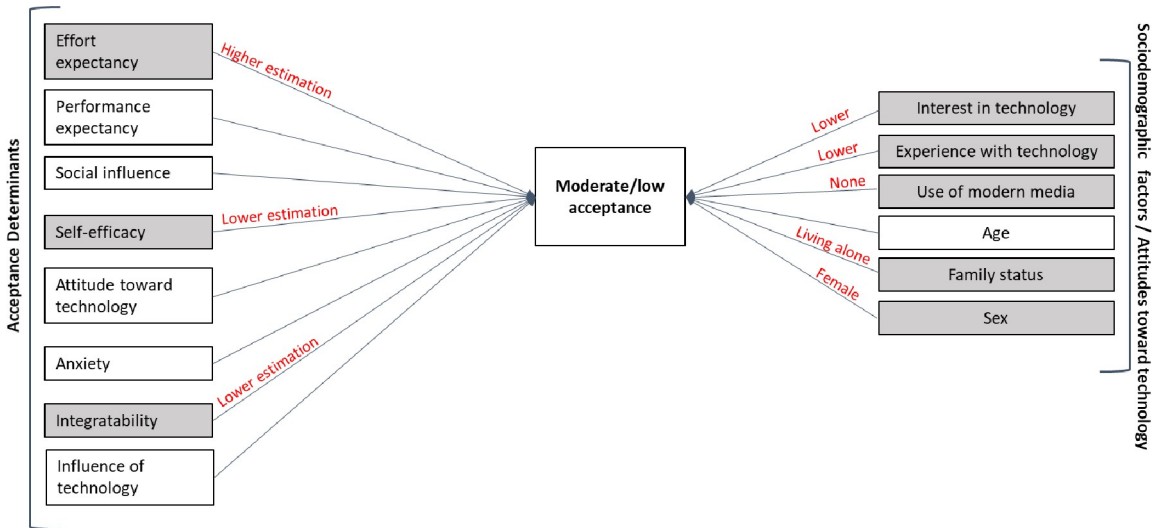

**Fig 2. Graphic illustration of the identified characteristics of the surveyed study patients with moderate/low acceptance (n = 3).**

bathroom. Mentioned as possible locations in which to integrate a camera were the television, computer or a mirror; each of them by one-third of the respondents.

### c) Patient concerns

A third (n = 6) of the respondents expressed concerns about the use of such a TMA. The reasons for concern were cited as being: fear of constant surveillance, a strange feeling caused by the camera, influences on the measurements caused by skin changes (e.g. in the presence of sun burn), data privacy, and operating the technology. When asked specifically, one respondent said that he could imagine feeling disturbed by the constant measuring and that he would be constantly reminded that he is not healthy.

## Discussion

In addition to improved disease management, the potential benefit of using TMAs in heart failure patients lies in the prevention of heart failure-related hospitalizations and decreased mortality. This potential can only manifest itself if the TMAs are accepted and utilized by the users. Potential users should therefore be included during development in order to achieve a high level of acceptance from the beginning.

The results of the acceptance survey presented here show that the respondents' acceptance of a new type of TMA (contactless camera-based measuring technology using iPPG) is already a priori quite high (83%). These respondents saw, for instance, a clear expected benefit in using such a TMA and that it could be integrated well into a daily routine.

Three of the respondents were rather skeptical of using this measuring technology. This involved mostly women (n = 2) living alone with little technological experience and little interest in technology. This lines up with the results of Lang et al., which found that older women living alone had more problems dealing with telemonitoring application [23]. These respondents also estimated the expected effort to use this TMA as higher, and its integratability and their self-efficacy as lower, results that were given special attention when identifying acceptance-enhancing measures and design suggestions.

Performance expectancy and effort expectancy, two determinants which, as in Dockweiler's study on mHealth apps in young people [24], strongly influence user acceptance also in this age group (60+). Acceptance sinks here, too, with a higher expectation of effort and/or a lower expectation of performance.

Also, as described by Dockweiler [11], there was a noticeable empathetic sensitivity on the part of the respondents for the other users (primary care physicians) who could be involved in the use of the TMA. The respondents were in part aware of the time and effort it entailed for their physicians, and they would use the TMA if their physician desired that they do so. At this point it should be noted that at the time of the survey, the new decision "Methods of Contract Medical Care Directive: Telemonitoring in heart failure" introduce in March 2021 was not available [9]. This decision (enter into force March 2021) provides for cooperation between telemedical centers and resident doctors. This distributes the effort and workload.

All of those surveyed, even those with low acceptance, were able to imagine that friends, family and/or their primary care physician would recommend this new measuring technology to them (social influence determinant). This is a very important aspect that is intrinsic to developing a positive attitude (and one that would otherwise be difficult to influence).

The concerns (e.g., fear of constant surveillance, constantly being reminded of disease) are indications that TMAs should always be used in a needs-oriented manner and that they are not suitable for everyone. Usage must be weighed critically for different target groups because TMAs, like this planned one, can also intensify the perception of disease. It should be

emphasized for the users that the constant observation, which does in a way fall under a definition of monitoring, brings with it a kind of empowerment and is not an instrument of control.

In this study the people who responded most openly to the planned TMA were already quite technophile and used modern technology (e.g. smartphone, tablet, computer), making it that much more important when developing a telemedical application to pay attention to the identified suggestions in order to increase the acceptance specifically of single women without technological experience.

### Identifying acceptance-enhancing measures/design suggestions

The following acceptance-enhancing measures and design suggestions were drawn from the determinant attributes for Group 2 (high effort expectancy, low integrability, low self-efficacy). The amount of time needed should be kept as low as possible through simple and easy operation. This also increases integrability into the daily routine. In addition to the presence of an integrated help or support function, which was felt to be very important, age-appropriate instructions should also be given prior to use in order to increase self-efficacy.

In regard to the ideas about designing a potential TMA, it should be noted that:

- The camera should be switched on and off by the users to take the measurements (this decreases concerns about constant surveillance);

- Direct feedback (with a choice between acoustic or visual) should be given regarding the measured vital parameters;

- Direct transmission of the measured values to the primary care physician should be possible;

- There should be an option to choose where the measuring device or camera is located (tablet, computer, television, mirror).

The final point (selecting the camera location) can also lead to better integration into the daily routine since the measurement can be taken in a place that is already routinely used by the patient.

The second to last point indicates that implementation should involve a combination of a store-and-forward and a real-time telemedicine application that enables not only the storage of data on the end device and weekly reports, but also the reporting of acutely abnormal measurements directly to the primary care physician.

### Limitations

The ability to draw conclusions on the basis of the results is limited due to the small sample size (especially the group 2 (n = 3) with moderate/low acceptance), which should be taken into account when interpreting the results. Despite this, the results give important indications for the development of the TMA, but they do not allow for any conclusions about actual later use.

A selection bias regarding the study patients cannot be ruled out. On one hand, there could be heightened interest; on the other, participation in the project could have had an influence that, in turn, could have exerted influence on acceptance. Surveying other people not participating in the project, however, would have become problematic because the survey referred to a very new measuring technology that requires a lot of explaining to people in this age group. Another possible limitation results from the translation and adaptation of the questions on the individual acceptance dimensions of the original UTAUT.

A six-point Likert scale was used in an attempt to avoid the tendency to choose the middle value when answering questions. It cannot be ruled out that responses to the qualitative

questions potentially reflect a certain social desirability. The heterogeneous data distribution suggests that this effect was mostly minimal.

## Outlook

A new generation of medical technology is emerging in the form of monitoring based on contactless measuring technology to determine vital parameters from video data. As a direct consequence of the current corona pandemic, contactless camera-based measuring technologies and telemonitoring applications will presumably gain even more in importance over the coming years, mainly for at-risk groups. This type of measuring technology not only offers new possibilities, but also raises new concerns on the part of users. Above all, these concerns need to be taken into consideration in order to develop feasible, accepted user-centered solutions. Because acceptance is an important prerequisite for adherence or adherent use. The results of this acceptance survey can serve as reference points for developing other telemonitoring applications.

It remains to be seen what results an acceptance survey would yield in the case of actual use (after successful development), or how adherent this use would be. For the latter, however, long-term analyses would be necessary.

## Conclusion

The acceptance of a new generation of medical technology (based on a contactless camera-based measuring technology) for a telemonitoring application is already quite high in the surveyed group of elderly users (60+). Users not only have specific ideas about the design, but also concerns that should be taken into consideration when developing the application in order to enable successful implementation.

## Supporting information

**S1 Fig. Questionnaire for patient survey.**
(PDF)

**S1 Table. Categorization of acceptance (regarding the intended use).**
(PDF)

**S2 Table. Categorization of the determinants.**
(PDF)

**S3 Table. Example of categorization for determinants with multiple items.**
(PDF)

**S4 Table. Inclusion and exclusion criteria for study participation.**
(DOCX)

**S1 File. Raw data.**
(XLSX)

## Acknowledgments

We wish to extend our thanks to all project partners for their constructive collaboration. Special thanks goes to our participating study patients and study physicians who, with their participation in the Home-based Healthcare Project and in the surveys and interviews, have contributed significantly to the study results.

## Author Contributions

**Conceptualization:** Peggy Borchers, David Pfisterer, Matthieu Scherpf, Karen Voigt, Antje Bergmann.

**Data curation:** Peggy Borchers, David Pfisterer.

**Formal analysis:** Peggy Borchers.

**Methodology:** Peggy Borchers, David Pfisterer, Karen Voigt.

**Writing – original draft:** Peggy Borchers.

**Writing – review & editing:** Peggy Borchers, David Pfisterer, Matthieu Scherpf, Karen Voigt, Antje Bergmann.

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
