## [Decision Letter · Decision Letter 0]

2 Oct 2022

PONE-D-22-18010Needs- and user-oriented development of contactless camera-based telemonitoring in heart disease – Results of an acceptance survey from the Home-based Healthcare Project (Feasibility Project)

PLOS ONE

Dear Dr. Borchers,

Thank you for submitting your manuscript to PLOS ONE. After careful consideration, we feel that it has merit but does not fully meet PLOS ONE’s publication criteria as it currently stands. Therefore, we invite you to submit a revised version of the manuscript that addresses the points raised during the review process.

It is requested to (see for details reviewers' comments):

- revise the conclusion/results presentation to stress the limitations due to low sample size;

- integrate/update literature review (e.g. including telemedicine in Hearth Failure);- address differentiation between non-invasive and invasive tele-monitoring;- specify selection criteria adopted for recruiting patients;- discuss limitations due to translation and reduction of items in the questionnaire;- present absolute values instead of relative ones in the results section;- discuss the time development of the survey in relation to the time development of the overall study;- list in the test the evaluation measurements taken during the survey;- revise the English language. 

A marked-up copy of your manuscript that highlights changes made to the original version. You should upload this as a separate file labeled 'Revised Manuscript with Track Changes'.An unmarked version of your revised paper without tracked changes. You should upload this as a separate file labeled 'Manuscript'.

We look forward to receiving your revised manuscript.

Kind regards,

Filomena Papa

Academic Editor

PLOS ONE

Journal Requirements:

Reviewers' comments:

Reviewer's Responses to Questions

**Comments to the Author**

1. Is the manuscript technically sound, and do the data support the conclusions?

Reviewer #1: Yes

Reviewer #2: Partly

2. Has the statistical analysis been performed appropriately and rigorously? 

Reviewer #1: Yes

Reviewer #2: Yes

3. Have the authors made all data underlying the findings in their manuscript fully available?

Reviewer #1: Yes

Reviewer #2: Yes

4. Is the manuscript presented in an intelligible fashion and written in standard English?

Reviewer #1: No

Reviewer #2: Yes

5. Review Comments to the Author

Reviewer #1: The paper explores user acceptance of contactless camera-based telemonitoring in heart disease

patients, as a part of The Home-based Healthcare Project.

I found it interesting to read, well written (although language fluency should be improved) and coherent.

The small sample size is not a problem in my opinion: it is a feasibility study and the "Limitations" section properly addresses the issue.

Inference statistical methods cannot be used due to the small sample size: the authors understand that and give themselves self-limitations in the scope of the paper.

Before publication, I suggest a few minor revisions:

* In the Development of the Questionnaire" section, I would not write that "According to which, the acceptance of a technology is expressed in the use of that technology". The original UTAUT paper by Venkatesh is already centered on the intention to use.

* Reference 18 (Venkatesh et al., 2003) should be complemented by more recent work by Venkatesh, expanding the Unified Theory (see, for example, Venkatesh, V., J. Thong, and X. Xu. 2016. “Unified Theory of Acceptance and Use of Technology: A Synthesis and the Road Ahead.” Journal of the Association for Information Systems 17 (5): 328–376)

* The evaluation measurements taken during the study should be listed in the text. Blood pressure is mentioned somewhere, but I would like to read the full catalogue.

* I suggest a thorough language revision. Sentences like "The measures to boost acceptance paid special attention this" in the abstract need adjustments.

Reviewer #2: Dear authors,

Thank you for reviewing your article. The paper covers an interesting topic but is missing relevant information like new literature and a discussion reflecting the overall limitations due to low sample size.

The low sample size is fine for feasibility study but the conclusion/results are presented in a way that it seems to be absolute. Presenting conclusion resulting of 3 patients (group 2) should be in a more descriptive/qualitative manner.

Further I give my comments for specific sections:

Introduction:

Actual literature is missing (ESC guidelines, TIM-HF2, OSICAT, German decision about reimbursement). Especially telemedicine in Heart Failure is a very fast developing research topic and has provided a lot of new research content within the last year. As the authors are from Germany they should reflect the decision of the GEmeinsamen Bundesausschuss to pay for telemedicine in HF as part of standard care (see decision process starting in December 2020 until Qualitätssicherungsvereinbarung in March 2022). The introduction also not differentiate between non-invasive and invasive telemonitoring; at least in one sentence this differentation should be addressed.

In the further paper Heart failure is not mentioned any more – are the survey respondents HF patients?

Please also provide a definition of “elderly”.

Methods

As this feasibility analysis was part of a overall study – has the overall study a registration number (e.g. DRKS)?

Line 85: Why written informed consent is in Brackets. Written informed consent is crucial for any study (see Good Clinical Practice). Also a statement to be in line with declaration on Helsinki is missing.

Why only 19 patient were recruited, when 20 were planned. The authors names “pragmatic reasons” for case low case number. Did they looking for patients regarding specific inclusion and exclusion criteria? Please specify, as this could be big bias.

The UTAUT is only valid in the English and only for the complete version. Please discuss a possible limitation due to translation and reduction of questions in the limitation sections.

Line 97: Are results of the study available. Which 250 measurements were performed?

Section data collection

The first sentence is not necessary. Why the survey ends (December 2020) before the study (July 2021)? Please specify

Line 158/159: As the case number is very low in the results section it is better to present absolute instead of relative numbers. As group 2 included only 3 persons the results are not comparable. This need to be discussed.

Why 3 groups were grouped when only 2 were presented? What are the (mean) values to be categorized in each group? Is the group “moderate/low” a mix of this 2 groups? According the results in Table 2 nobody was categorized as “low acceptance”. Why was the column then named in this way?

Results

Why only 18 patients participated in the evaluation survey? How long did the patients use the measuring system when they answered the questionnaire. Why the questionnaire was not used after the overall study was finished?

Patients idea: visual and acoustic combined – why is this the same group as acoustic? Were Multiple answers possible?

What means “followed by the study” (line 226)?

In the results section results of both groups are often presented together. I would suggest to do this for all results and not to provide to groups. Difference in the groups could be discussed qualitative; but due to the n=18 vs. n=3 patients in groups these results are not comparable. For this reason figure 2 cannot be so conclusive; please name it in a way like “possible characteristics”. When a sample only consists of 3 person, all results can be random.

Line 236: How sun burn can be happen due to use of the camera?

Line 252: “This involved mostly women…” As 17% of 18 participant are only 3 persons, 2 persons have to be female. A conclusion in this way is not correct. The discussion should present absolute numbers to reflect this bias of a low case number.

TMA structure for HF patients (personal, technical) is actually clearly defined by the G-BA; reimbursement is possible in Germany (decision in March 2021, reimbursement numbers defined in January 2022). The decision was not done as the survey was performed but it should be discussed as e.g. a fear of overwhelming the primary care physician is not longer necessary. Physician can decided whether to perform TMA itself or by a telemedical centre.

Line 281: What means modern technology?

Also, the authors could discuss that acceptance does not mean compliance or adherence. A long term analysis is needed for this.

6. PLOS authors have the option to publish the peer review history of their article (what does this mean?). If published, this will include your full peer review and any attached files.

Reviewer #1: **Yes: **Bartolomeo Sapio

Reviewer #2: No

---

## [Author Response · Author response to Decision Letter 0]

9 Jan 2023

Dear Editor,

Thank you for considering our manuscript for publication. We are happy to revise this according to the reviewer comments. However, I have a question: A reviewer criticized the language. This point is quite general. The manuscript was officially commissioned for translation and was translated by a state-certified translator. Since only one reviewer noted the language, I would like to ask you, the editor, for your opinion as to whether an improvement is necessary.

Best regards, on behalf of all authors

Peggy Borchers

The responses to the reviewers' comments have been uploaded as a separate document.

---

## [Decision Letter · Decision Letter 1]

24 Jan 2023

PONE-D-22-18010R1Needs- and user-oriented development of contactless camera-based telemonitoring in heart disease – Results of an acceptance survey from the Home-based Healthcare Project (Feasibility Project)PLOS ONE

Dear Dr. Borchers,

Thank you for submitting your manuscript to PLOS ONE. After careful consideration, we feel that it has merit but does not fully meet PLOS ONE’s publication criteria as it currently stands. Therefore, we invite you to submit a revised version of the manuscript that addresses the points raised during the review process.

It is requested to:

- revise the manuscript addressing comments from Reviewer 2;

- include in the manuscript (or in the supporting information) items utilized for measurement of all determinants (in English);

- copy edit the manuscript correcting possible errors, for instance a the end of Table 1 eliminate “Error! Reference source not found”.

We look forward to receiving your revised manuscript.

Kind regards,

Filomena Papa

Academic Editor

PLOS ONE

Journal Requirements:

Additional Editor Comments (if provided):

  Answer to the authors' question about translation: since Reviewer1 approved Revision 1 it is not necessary to improve translation.

However it is requested to:

-include in the manuscript (or in the supporting information) items utilized for measurement of all determinants (in English);

- copy edit the manuscript correcting errors (if any), for instance a the end of Table 1 eliminate “(Error! Reference source not found)”.

Reviewers' comments:

Reviewer's Responses to Questions

**Comments to the Author**

1. If the authors have adequately addressed your comments raised in a previous round of review and you feel that this manuscript is now acceptable for publication, you may indicate that here to bypass the “Comments to the Author” section, enter your conflict of interest statement in the “Confidential to Editor” section, and submit your "Accept" recommendation.

Reviewer #1: All comments have been addressed

Reviewer #2: (No Response)

2. Is the manuscript technically sound, and do the data support the conclusions?

Reviewer #1: Yes

Reviewer #2: Yes

3. Has the statistical analysis been performed appropriately and rigorously? 

Reviewer #1: Yes

Reviewer #2: Yes

4. Have the authors made all data underlying the findings in their manuscript fully available?

Reviewer #1: Yes

Reviewer #2: Yes

5. Is the manuscript presented in an intelligible fashion and written in standard English?

Reviewer #1: Yes

Reviewer #2: Yes

6. Review Comments to the Author

Reviewer #1: I am satisfied with the revision proposed by the authors.

The paper can now be published in the journal.

Reviewer #2: Dear authors,

thank you for taking note of my recommendations. I still do not understand the evaluation concept completly. You are writing that 250 evaluation measurements were taken by the study nurses in the patients house (methods sections).

In your review comments you mentioned that „an average of 5 evaluation measurements were carried out before the interview.“. Are these measurements taken by the patients themself? (otherwise 250 measurement divided by 19 patients does not average 5 measurements). Also you write, that the evaluation was taken by phone. Why was the survey not carried out directly after the measurements by the study nurse? How long was the time interval between measurement and evaluation survey?

Many thanks in advance for answering my questions.

Also a note for a further reference for your introduction section: https://academic.oup.com/eurjcn/advance-article/doi/10.1093/eurjcn/zvac080/6691856

7. PLOS authors have the option to publish the peer review history of their article (what does this mean?). If published, this will include your full peer review and any attached files.

Reviewer #1: **Yes: **Bartolomeo Sapio

Reviewer #2: No

---

## [Author Response · Author response to Decision Letter 1]

15 Feb 2023

The responses to the reviewers' comments have been uploaded as a separate document ("Response to Reviewer").

---

## [Decision Letter · Decision Letter 2]

17 Feb 2023

Needs- and user-oriented development of contactless camera-based telemonitoring in heart disease – Results of an acceptance survey from the Home-based Healthcare Project (Feasibility Project)

PONE-D-22-18010R2

Dear Dr. Borchers,

We’re pleased to inform you that your manuscript has been judged scientifically suitable for publication and will be formally accepted for publication once it meets all outstanding technical requirements.

Kind regards,

Filomena Papa

Academic Editor

PLOS ONE

Additional Editor Comments (optional):

Reviewers' comments:

Reviewer's Responses to Questions

**Comments to the Author**

1. If the authors have adequately addressed your comments raised in a previous round of review and you feel that this manuscript is now acceptable for publication, you may indicate that here to bypass the “Comments to the Author” section, enter your conflict of interest statement in the “Confidential to Editor” section, and submit your "Accept" recommendation.

Reviewer #1: All comments have been addressed

Reviewer #2: All comments have been addressed

2. Is the manuscript technically sound, and do the data support the conclusions?

Reviewer #1: Yes

Reviewer #2: Yes

3. Has the statistical analysis been performed appropriately and rigorously? 

Reviewer #1: Yes

Reviewer #2: Yes

4. Have the authors made all data underlying the findings in their manuscript fully available?

Reviewer #1: Yes

Reviewer #2: Yes

5. Is the manuscript presented in an intelligible fashion and written in standard English?

Reviewer #1: Yes

Reviewer #2: Yes

6. Review Comments to the Author

Reviewer #1: I am satisfied with the revision proposed by the authors.

The paper can now be published in the journal.

Reviewer #2: (No Response)

7. PLOS authors have the option to publish the peer review history of their article (what does this mean?). If published, this will include your full peer review and any attached files.

Reviewer #1: **Yes: **Bartolomeo Sapio

Reviewer #2: No

---

## [Editor Report · Acceptance letter]

21 Feb 2023

PONE-D-22-18010R2 

Needs- and user-oriented development of contactless camera-based telemonitoring in heart disease – Results of an acceptance survey from the Home-based Healthcare Project (Feasibility Project) 

Dear Dr. Borchers:

I'm pleased to inform you that your manuscript has been deemed suitable for publication in PLOS ONE. Congratulations! Your manuscript is now with our production department. 

Kind regards, 

on behalf of

Dr. Filomena Papa 

Academic Editor

PLOS ONE